# The origin of enhanced $O_2^+$ production from photoionized $CO_2$ clusters

Smita Ganguly [1], Dario Barreiro-Lage [2], Noelle Walsh[3], Bart Oostenrijk [1], Stacey L. Sorensen [1], Sergio Díaz-Tendero [2,4,5✉] & Mathieu Gisselbrecht [1✉]

$CO_2$-rich planetary atmospheres are continuously exposed to ionising radiation driving major photochemical processes. In the Martian atmosphere, $CO_2$ clusters are predicted to exist at high altitudes motivating a deeper understanding of their photochemistry. In this joint experimental-theoretical study, we investigate the photoreactions of $CO_2$ clusters ($\leq 2$ nm) induced by soft X-ray ionisation. We observe dramatically enhanced production of $O_2^+$ from photoionized $CO_2$ clusters compared to the case of the isolated molecule and identify two relevant reactions. Using quantum chemistry calculations and multi-coincidence mass spectrometry, we pinpoint the origin of this enhancement: A size-dependent structural transition of the clusters from a covalently bonded arrangement to a weakly bonded polyhedral geometry that activates an exothermic reaction producing $O_2^+$. Our results unambiguously demonstrate that the photochemistry of small clusters/particles will likely have a strong influence on the ion balance in atmospheres.

[1] Department of Physics, Lund University, Box 118, SE-221 00 Lund, Sweden. [2] Departamento de Química - Módulo 13, Universidad Autónoma de Madrid, 28049 Madrid, Spain. [3] MAXIV laboratory, Lund University, Box 118, SE-221 00 Lund, Sweden. [4] Condensed Matter Physics Center (IFIMAC), Universidad Autónoma de Madrid, 28049 Madrid, Spain. [5] Institute for Advanced Research in Chemical Sciences (IAdChem), Universidad Autónoma de Madrid, 28049 Madrid, Spain. ✉email: sergio.diaztendero@uam.es; mathieu.gisselbrecht@sljus.lu.se

Molecular clusters have distinctive physicochemical properties that evolve from free molecules to bulk-like with increasing cluster size. The presence of inter-molecular interactions within these clusters creates a rich environment for the complex interplay between geometry and intra-cluster chemistry. In planetary atmospheres, molecular clusters act as precursors to larger particles like aerosols and cloud nuclei[1–5] and the interaction of solar radiation with these atmospheric particles drives both the chemistry and radiative transfer mechanisms that control planetary surface temperatures and climate.

In $CO_2$-rich atmospheres like that of Mars, the presence of $CO_2$ particles at different altitudes between 60 and 100 km has been confirmed in the form of $CO_2$ ice aerosols[6] and $CO_2$ ice clouds[7–10]. However, the exact composition of these particles is the subject of debate. Whilst classical nucleation theories rule out homogeneous nucleation of $CO_2$ molecules into larger particles in the atmosphere of Mars[11], recent quantum chemical calculations predict the existence of pure $CO_2$ clusters at high altitudes[12].

Models for predicting and understanding the Martian atmosphere are based upon photochemical processes aimed at predicting the concentration of the most abundant charged/neutral species. Observations from the many space probes which have passed or landed on Mars provide information on atmospheric constituents and can be compared to these models. Cardnell et al. recently revealed the fundamental role that aerosols and water clusters play in the photochemistry that takes place at altitudes <70 km on Mars[13,14] and the influence of molecules at altitudes >100 km in the Martian atmosphere is well understood[15,16]. However, the photochemistry of small $CO_2$ clusters in the interface between these extremes is not fully elucidated.

Ionospheric models and observations show that $O_2^+$ is the dominant ionic species in the lower Martian ionosphere.[14,15,17] At these altitudes, $O_2$ is believed to be primarily produced by photodissociation of $CO_2$ molecules[18]. However, the $O_2$ densities calculated by Lo et al.[19] are an order of magnitude less than the direct measurements indicating the presence of additional $O_2$ sources. Therefore, to understand the mechanisms that lead to such significant $O_2^+$ production in a $CO_2$-rich environment, it is necessary to consider all photoreactions that may take place at these altitudes.

The photochemistry that occurs at altitudes >70 km is significantly influenced by solar soft X-rays[20] as these can penetrate deeper into the atmosphere and ionise atmospheric particles (e.g. $CO_2$ clusters). When a soft X-ray photon of sufficient energy ionises a molecular cluster, an inner-shell electron can be ejected. The excited cluster subsequently relaxes via Auger decay, resulting in the formation of a multiply-charged cluster ion[21]. The stability of the cluster ion depends mainly upon its size[22], and clusters smaller than a critical size will dissociate in order to reach energetic stability[23]. Previous studies have demonstrated that the dissociation of small multiply charged $CO_2$ cluster ions can produce $O_2^+$[24–27] and to date, the production of $O_2^+$ from such ionised $CO_2$ clusters has been attributed to collision and recombination processes of $O^+$ with surrounding molecules in the cluster[24–26]. However, these processes are only superficially understood. Therefore, given the crucial role of $O_2^+$ in atmospheric chemistry a detailed study of the soft X-ray induced dissociation of $CO_2$ cluster ions into $O_2^+$ is well motivated.

In this work, we present new insights into the production of $O_2^+$ ions from $CO_2$ clusters, deduced with the help of a multi-coincidence mass spectrometer[28] and theoretical calculations. We observe that soft X-ray ionised $CO_2$ clusters consisting of a few to ~100 molecules produce a significant amount of $O_2^+$. For the range of cluster diameters (≤2 nm) in our study, ionisation is expected to mainly take place on the surface of the cluster. Our use of a position-sensitive detector at the end of the ion time-of-flight spectrometer enables us to perform 3D-momentum imaging of the ions produced in the cluster dissociation and we use this information to interpret the dynamics of the photoreactions. We found that multiple decay processes are accessible to unstable doubly charged clusters and studied two distinct mechanisms that lead to production of $O_2^+$; both reactions exhibit a strong dependence on the size of the precursor cluster. Ab initio quantum chemical calculations were used to determine structures for clusters, potential energy surfaces, and the structural evolution of the doubly-ionised clusters as a function of size. The theory provides a kinematic picture of the possible pathways accessible in an exciting cluster dication, but most significantly, our calculations identify fundamental changes in the structure of the cluster dications which are likely related to the emergence of the different mechanisms for producing $O_2^+$. Thus, here we report an enhancement in the yield of $O_2^+$ produced from ionised $CO_2$ clusters compared to the case of $O_2^+$ production from ionised $CO_2$ molecules[29] and our results highlight that the mechanism for producing $O_2^+$ is closely related to the geometric structure of the cluster dications. Our findings shed light on the importance of the photochemistry of even the smallest particles present in planetary atmospheres.

## Results

The principle of our measurements is presented in Fig. 1a and explained in more detail in the methods section. Briefly, X rays intersect a beam composed of a mixture of free neutral $CO_2$ molecules and clusters at the centre of a multi-coincidence 3D-momentum imaging mass spectrometer[28,30]. Measurements are only recorded upon detection of an electron, enabling detection of each cluster 'ionisation/fragmentation event' independently. Two ions detected during a single 'ionisation/fragmentation event' are identified as ions measured in coincidence. Figure 1b depicts a coincidence map of all ions detected following $(CO_2)_n^{2+}$ cluster fragmentation after ionisation at 320 eV (above the C1s edge). The mean cluster size in the beam, $N_{mean}$, was about 20. The relative intensities of the fragmentation channels resulting from the dissociation of the unstable multiply-charged clusters[21] are reflected in this plot. The most prominent experimentally observed fragmentation channels correspond to coincident detection of $(CO_2)_m^+/(CO_2)_n^+$ $(m, n = 1, 2, 3, . . .)$, i.e. related to the dissociation of clusters into smaller (singly-charged) units, in agreement with previous studies[21,31]. The next dominant fragmentation channel is associated with intra-cluster reactions related to $O_2^+$ production; these channels are of the type-$O_2^+/(CO_2)_m^+$ $(m = 1, 2, 3, . . .)$. The yield of $O_2^+$ production from clusters is clearly distinct from that observed following the photo dissociation of free $CO_2$ molecules at 320 eV. In the case of free doubly-charged $CO_2$ molecules, bond rearrangement before fragmentation leads to the production of the $O_2^+/C^+$ pair, and the yield of $O_2^+$ from this process is below 1% at this photon energy[29]. The pie chart in Fig. 1c clearly shows that the yield of $O_2^+/(CO_2)_m^+$ channels in clusters is significantly larger than the molecular $O_2^+/C^+$ channel, despite an ~10 times higher fraction of isolated molecules in the beam (see Supplementary Table 1). The strong enhancement of $O_2^+$ production clearly originates from the clusters rather than from the molecular portion of the beam.

To understand the dynamics of the photochemical reactions leading to this enhancement, we analysed a subset of our data, namely the $O_2^+/CO_2^+$ channel (the dominant channel in the pie chart in Fig. 1c). The imaging capability of our spectrometer allows the 3D momenta of individual ions to be determined and hence, the momentum correlation between ionic fragments can be mapped. The dissociaton of these doubly-charged clusters may also produce additional neutral fragments which cannot be

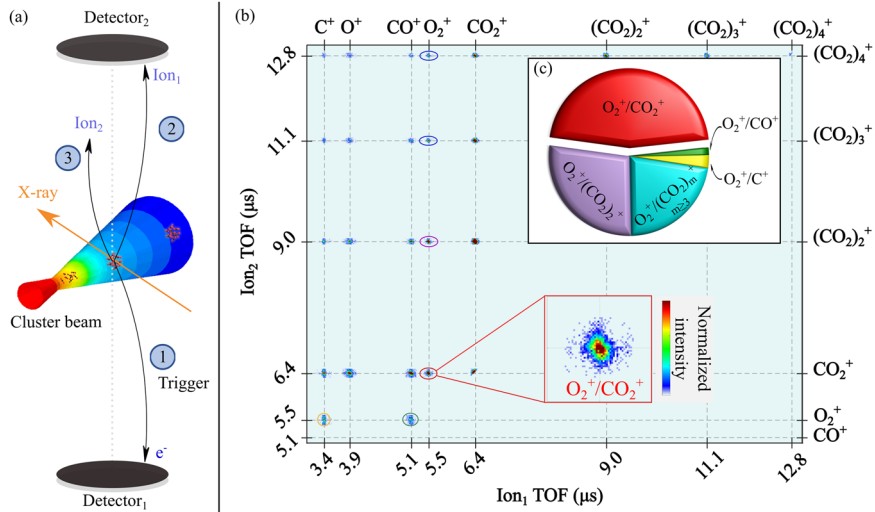

**Fig. 1 Principle of the measurement. a** Schematic of the experiment showing a typical double-ion coincidence measurement. After ionisation, the detection of an electron, Detector₁, triggers the acquisition system and the arrival time-of-flight (TOF) of the ions and their positions are recorded on Detector₂, allowing us to determine the momentum of individual ions. **b** Two-dimensional coincidence map of Ion₁ TOF vs Ion₂ TOF produced following ionisation of $CO_2$ clusters ($N_{mean}$ ~ 20) at 320 eV. The data is filtered to remove false coincidences. The cluster dissociation channels are shown, together with an expanded view of the $O_2^+/CO_2^+$ channel. **c** Pie chart indicating the dissociation channels that lead to $O_2^+$ production from clusters ($N_{mean}$ ~ 20) along with the $O_2^+/C^+$ channel. In free $CO_2$ molecules, $O_2^+/C^+$ is the only channel that can produce $O_2^+$, however, in clusters this channel is overshadowed by the dominating $O_2^+/(CO_2)_m^+$ channels ($m = 1, 2, \ldots, 10$).

detected by our system. However, the momenta of these fragments are encoded in the kinematics of the detected ions and can be extracted using the following approach: The momentum of the undetected cluster fragments is defined as $\overrightarrow{P_{res}} = -\sum_j \overrightarrow{P_j}$ where $j$ denotes the measured ions and by using a framework for a three-body break up, the momentum correlation between the measured ions and the undetected particles can be visualised using a *Dalitz plot*[32,33].

Figure 2 shows the evolution of the Dalitz plot for two different average cluster sizes, $N_{mean}$ ~4 and ~20. The coordinates are given by $\epsilon_i = |\vec{P_i}|^2 / \sum_i |\vec{P_i}|^2$ for each particle $i$. For small cluster sizes, $N_{mean}$ ~4, a distribution can be observed on the left-hand side of the plot where the momenta of $O_2^+$ and of the residual cluster fragments are strongly anti-correlated; we refer to this as region A (highlighted in red in Fig. 2). The fragments associated with this region are produced sequentially as follows:

$$(CO_2)_k^{2+} \rightarrow T^+ + CO_2^+ \rightarrow \ldots \rightarrow X + O_2^+ + CO_2^+ \quad (1)$$

where $T^+$ represents a short-lived transient species that undergoes fragmentation. As the $CO_2^+$ ion is produced first, its momentum is uncorrelated to $O_2^+$ and the undetected residual cluster fragments, $X$.

For slightly larger cluster sizes ($N_{mean}$ ~20), a broad distribution appears on the right-hand side of the Dalitz plot; we denote this region B (highlighted in blue in Fig. 2). A continuous distribution that extends between regions A and B is also evident in the plot. For fragments associated with region B, the momentum of $CO_2^+$ and of the residual cluster fragments are strongly anti-correlated and the momentum of $O_2^+$ is uncorrelated. This corresponds to a sequential fragmentation process

$$(CO_2)_k^{2+} \rightarrow T^+ + O_2^+ \rightarrow \ldots \rightarrow X + CO_2^+ + O_2^+. \quad (2)$$

The analysis of the coincidence data using the Dalitz plot provides strong evidence that the yield of $O_2^+$ arises mainly due to sequential fragmentation. In one case, leading to prompt ejection of $O_2^+$, in the other case with a delay. The Dalitz plot for the $O_2^+/(CO_2)_2^+$ channel also shows prompt emission of $O_2^+$ (see Supplementary Figure 3). The previously proposed mechanism in the literature[24–26] for the formation of $(CO_2)_m O_2^+$ species,

invoking collision and recombination of $O^+$ with surrounding molecules in the cluster, is not sufficient to explain the signatures of multiple processes observed in the Dalitz plots.

To investigate the competition between the multiple-fragmentation channels, we look at the ratio of measured events in regions A and B in the Dalitz plots that were generated for different mean cluster sizes. Figure 2b shows the cluster-size dependence of this ratio. A small cluster sizes region A is dominant, indicating a preference for sequential breakup with a delayed emission of $O_2^+$. As the cluster size increases to $N_{mean}$ ~11, region B becomes dominant in the Dalitz plot, favouring a sequential breakup up with prompt emission of $O_2^+$. Thus, we conclude that the fragmentation channel contributing to region B is sensitive to the size of the parent cluster and we find that it is only present when $N_{mean} > 4$. The results of our quantum chemical calculations (see Methods) across a wide range of cluster sizes indicates a structural transition around size $N = 11$. We find that doubly-charged clusters with only a few molecules form a covalently-bonded species of a specific structure, which we subsequently name "2Y-structures". In contrast doubly charged clusters composed of more than 12 molecules stabilise as closed-shell polyhedral structures (see Supplementary Note 3). Therefore, our results indicate that the fragmentation channel contributing to region B in the Dalitz plot is activated when the closed-shell polyhedral structure is formed.

To gain deeper insight into the multiple reaction mechanisms leading to the formation of the $O_2^+/CO_2^+$ channel, we performed ab initio molecular dynamics (AIMD) simulations of the first few hundred femtoseconds after ionisation of a $(CO_2)_5^{2+}$—our test model. The simulations reveal that there are two different primary mechanisms leading to the formation of $O_2^+$ from doubly-ionised $(CO_2)_5^{2+}$ cluster. The first of these involves the formation of complex $(CO_2)_m^{2+}$ structures (see Fig. 3a), whilst the second mechanism is invoked after dissociation of a $CO_2$ molecule and subsequent intracluster reactions with surrounding $CO_2$ molecules (see Fig. 3b). Comparison of the yield ratio between different fragmentation channels as determined from our AIMD simulations is in agreement with our observations and thereby confirms the validity of our model. This leads us to an estimate of the

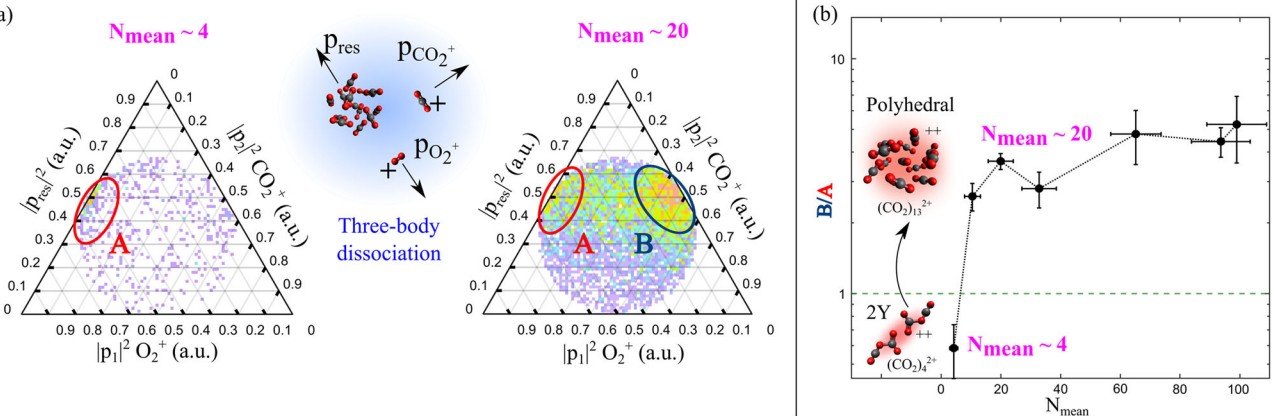

**Fig. 2 Sequential cluster dissociation processes forming $O_2^+$. a** Dalitz plot (guide for interpretation is provided in Supplementary Fig. 2) showing the momentum distribution between the fragments in the $O_2^+/CO_2^+$ channel for different mean cluster sizes ($N_{mean}$). In the centre, we schematically show the three-body dissociation of the mother cluster into $O_2^+$, $CO_2^+$ and the undetected residual cluster fragment. For smaller cluster sizes $N_{mean} \sim 4$, we observe a distribution labelled region A (in red, left-hand side of Dalitz plot), for larger cluster sizes a second distribution labelled region B (in blue) appears to the right. **b** The cluster-size dependence of the intensity ratio between region A and region B from the $O_2^+/CO_2^+$ channel. The error bars for the $N_{mean}$ values are calculated as described in Harnes et al[65]. The geometry of the doubly-charged cluster evolves from a 2Y-structure for small clusters to a polyhedral structure for cluster sizes above 11.

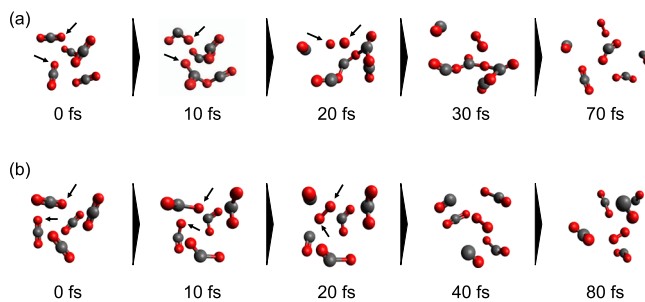

**Fig. 3 Snapshots of molecular dynamics trajectories.** Starting from a $(CO_2)_5^{2+}$ cluster leading to the formation of $CO_2^+ + O_2^+$ **a** through the formation of a covalently-bonded structure and **b** through $O^+/CO^+$. The arrow indicates the site of subsequent bond formation in each step.

internal energy remaining in the cluster after photoionisation - which is on the order of several tens of eVs (see Supplementary Note 6).

In order to unravel the energetic aspects of these mechanisms, we also explored the potential-energy surfaces (PES) involved in producing these fragments, thus identifying intermediate species and transition states in the paths that lead to the $O_2^+/CO_2^+$ exit channel (see Supplementary Note 4). When a neutral cluster is irradiated with X-rays, the ionisation of a molecule in the cluster creates a localised core–hole state. The singly-charged cluster will very likely emit a second electron to fill the inner-shell vacancy, and this creates a second vacancy in either the same molecule via Auger decay or in the neighbouring molecule for instance through Intermolecular Coulombic Decay[34–42]. This type of non-local Auger decay leads to rapid redistribution of the charge and of electronic excitation in the cluster $(CO_2)_m^{2+}$. In the upper panel of Fig. 4, we illustrate a mechanism starting from $(CO_2)_4^{2+}$—an example of a fragmentation pathway after charge and energy redistribution. The most favourable channel corresponds to the fragmentation of the cluster into $2CO_2^+ + 2CO_2$, in agreement with the experimental results. The PES indicates other pathways that are accessible in clusters with internal energy on the order of ~1–2 eV. In this case, stable structures corresponding to covalently bonded $(CO_2)_4^{2+}$ are formed. With additional internal energy, a variety of intra-cluster chemical reactions can be activated from this structure. We follow the pathways of these

reactions and find that they result in two endothermic reactions, asymptotically reaching the dissociation channels $O_2^+/CO_2^+$ and $O_2^+/CO^+$; both of these are observed in the experiments.

A local Auger decay creates a doubly ionised molecule within the cluster, $CO_2^{2+}$, which is surrounded by several neutral $CO_2$ molecules. The excess positive charge within a single molecule facilitates the molecular Coulomb Explosion, which creates $O^+$ and $CO^+$ fragments with substantial kinetic energy. The surrounding neutral molecules act as a cage, increasing the probability of recombination of $O^+$ in the cluster. This in turn results in the production of $O_2^{+43–45}$ through the formation of the $OCOO^+$ ion[46,47] or other more complex structures. Ultimately a charge-transfer process takes place between the $CO^+$ fragment and one of the molecules in the cage, producing $CO_2^{+48,49}$. The PES in the lower panel of Fig. 4 shows the path toward production of $O_2^+/CO_2^+$ in an exothermic process. Note that the charges of (metastable) $CO_2^{2+}$ can be delocalised due to interaction with surrounded molecules, opening the exothermic channel, which eventually results in smaller singly-charged cluster units.

## Discussion
Our experimental and theoretical results unambiguously indicate that the $O_2^+$ ion production in $CO_2$ clusters is the result of rich photochemistry, and includes multiple mechanisms which cannot be explained by the models proposed in the literature. To gauge the relative importance of these mechanisms, we investigate the endothermic and exothermic reactions as the cluster size grows.

In Fig. 5, the yields of $O_2^+$ in different channels relative to the $O_2^+$ yield in free molecules as a function of the cluster size are shown. All yields present a steep increase in the small cluster range, from ~4 to ~10 molecules. The yield of $O_2^+/CO^+$, which arises from an endothermic reaction with a high activation barrier, reaches a peak around 10–20 molecules, then decreases gradually to a constant level for larger clusters. This behaviour reflects the residual (internal) energy being redistributed as heat among a growing number of degrees of freedom as the cluster size increases. Above a critical size, this internal energy is not sufficient to pass the activation barrier, leading to the almost complete extinction of this channel.

The total $O_2^+/CO_2^+$ yield exhibits a peaked structure with contributions from different types of reactions. Similar to the $O_2^+/CO^+$ channel, an endothermic reaction can be expected as

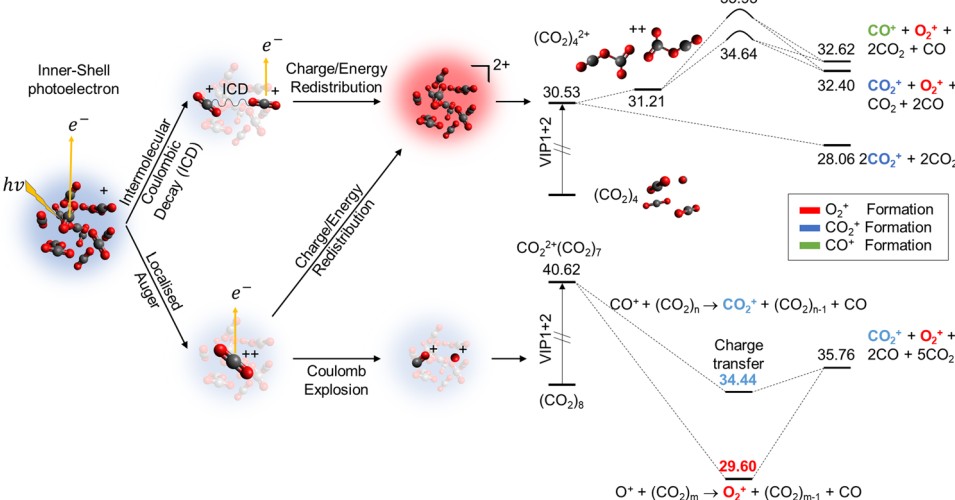

**Fig. 4 Possible paths that a $(CO_2)_n$ cluster can follow after core-electron ionisation followed by Auger emission.** (i) Charge and internal energy are redistributed along the cluster, which leads to the formation of stable $(CO_2)_4^{2+}$—2Y structures—in the potential energy surface. (ii) Coulomb explosion of a single molecule $CO_2^{2+} \rightarrow O^+ + CO^+$, which leads to the reaction of both charged species with the surrounding $CO_2$ molecules. Relevant points in the potential energy surface leading to the most important channels are shown, energy values are given in eV and refereed to the neutral cluster.

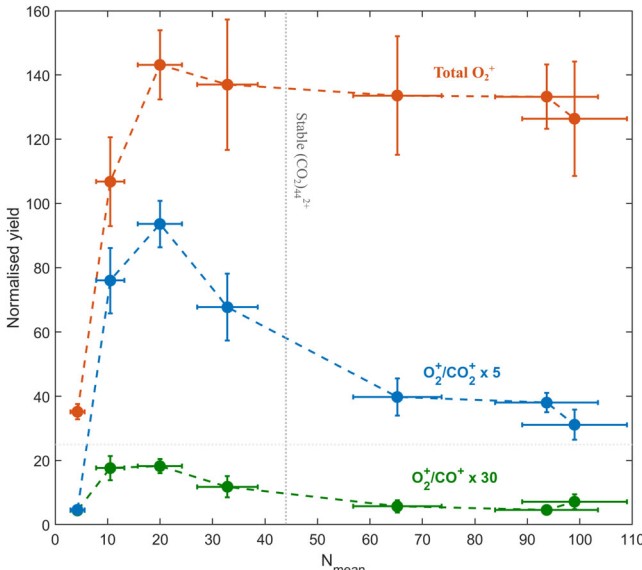

**Fig. 5 Relative increase of $O_2^+$ yield from $CO_2$ clusters with reference to $CO_2$ molecules.** Experimental $O_2^+$ yields as a function of the mean cluster size ($N_{mean}$) for events in coincidence with $CO^+$ ion (green) or $CO_2^+$ ion (blue), and for all $O_2^+$ events (red) including single ion events. The yields are normalised to the $O_2^+$ yield from the molecule given by the coincidence signal $O_2^+/C^+$. The dotted line shows the formation of a stable doubly-charged cluster above the critical size[22] of $N_{mean} = 44$.

The total yield of $O_2^+$ behaves essentially as a step function, with a slight maximum between $N_{mean}$ ~10 and ~20 molecules. This behaviour indicates that there is a minimum size required to activate the exothermic reaction. The efficiency of the reaction is then roughly constant for large clusters, even beyond the critical size for the formation of stable doubly-charged species.

In summary, we observe that the interaction of soft X-ray radiation with $CO_2$ clusters results in an enhanced yield of $O_2^+$ compared to the case of free molecules. We specifically investigate: (i) a sequential fragmentation process that involves prompt emission of $CO_2^+$, followed by $O_2^+$ emission, and (ii) the opposite sequence—initial loss of $O_2^+$ followed by loss of $CO_2^+$. The relative importance of these two processes is observed to depend upon cluster size and our theoretical modelling indicates that a structural transition of the cluster dication from the 2Y shaped covalently bonded structure to the polyhedral shaped weakly bonded structure is the origin of this dependence. Using quantum chemistry calculations we explored the pathways that lead to $O_2^+$ production and compared this to our experimental $O_2^+$ yield. The results reveal that an exothermic reaction is relevant for the case of localised charged states of the cluster. In contrast, for delocalised charged states we find that two endothermic reactions play a role.

In $CO_2$ rich atmospheres, when the conditions are favourable for $CO_2$ cluster nucleation, these clusters may strongly influence the photochemistry that takes place. Our results demonstrate that the $O_2^+$ yield will be at least 100 times enhanced compared to the case when only uncondensed $CO_2$ molecules are present in the atmosphere. In analogy to the Martian photochemical models used to simulate the behaviour of water clusters at low altitudes[14], the work presented here highlights the need to consider the potential influence of $CO_2$ clusters at higher altitudes.

predicted by our theoretical results. The significant yield of $O_2^+$ that remains for the larger clusters is the signature of an exothermic reaction due to intra-cluster collisions. A detailed study of the exothermic reaction is beyond the scope of this paper. As the mean cluster size increases, a multitude of competing fragmentation channels involving $(CO_2)_m^+$ ($m \geq 3$) units become accessible. However, the yield of these channels involving larger fragments in coincidence with $O_2^+$ is limited by the experimental detection efficiency and can result in *aborted coincidence* events[50] with only $O_2^+$ measured. Therefore, in Fig. 5 we present the total yield of $O_2^+$ to account for all possible fragmentation channels.

## Methods

The experiments were performed at the soft X-ray beamline I411 at MAX-lab, Lund, Sweden. We used a multi-ion coincidence 3D momentum imaging spectrometer described in Laksman et al.[28] and a specially developed data analysis package for clusters[30,51]. $CO_2$ clusters were produced using supersonic nozzle expansion of gaseous $CO_2$. The mean cluster size in the beam was estimated using the $\Gamma^*$ formalism (see Supplementary Note 1). The photon energy was kept constant at 320 eV, about 20 eV above the C 1s ionisation edge. Since we study the photodissociation of the doubly-charged cluster ion, only double coincidence data

with two ions detected simultaneously was used. The ionisation probability is assumed to be a Poisson distribution, the statistical error for sample size N is estimated to be $\sqrt{N}$. Further information about the experimental technique and data processing can be found in Supplementary Note 2.

The Dalitz plots at different cluster sizes were filtered to plot the ratios of a number of events in regions A and B in Fig. 2b. The kinetic energy (KE) filters used to define regions A and B

- Region A: $KE_{O_2^+} \leq 20.99$ eV and $KE_{CO_2^+} \leq 0.15$ eV
- Region B: $KE_{O_2^+} \leq 0.37$ eV and $KE_{CO_2^+} \leq 15.27$ eV

From the theory side, neutral structures up to 13 $CO_2$ molecules were taken from Takeuchi's work[52] and re-optimised using density functional theory (DFT), in particular with the Minnesota functional M06-2X[53] and the Pople basis set 6-31++G(d)[54–56]. Vertical Ionisation Potentials, cationic and dicationic structures and the PES exploration for the most interesting channels were obtained at the same level of theory. We have checked the accuracy of the relative energies computed at this level of theory by comparing them with results obtained at a much higher level of theory, namely DLPNO-CCSD(T)/def2-TZVP[57–59], for one of the computed mechanisms (see Supplementary Note 4). In order to study the temporal evolution of the doubly-ionised excited clusters, ab initio molecular dynamics simulations were performed using the Atom-centred Density Matrix Propagation (ADMP) formalism[60–62]. Trajectories have been carried out at a DFT level of theory, also using the M06-2X functional with the 6-31++G(d) basis set. All simulations were performed with a time step of $\Delta t = 0.1$ fs, a fictitious mass of $\mu = 0.1$ a.u., and a maximum propagation time of $t_{max} = 200$ fs. A typical excess/excitation internal energy, $E_{exc} = 25$, 40 and 50 eV, was used to consider states prepared upon ionisation. For each value of excitation energy, 100 trajectories have been carried out. All the calculations were performed using the Gaussian16 programme[63], except the CCSD(T) ones that were carried out using Orca 5.0 programme[64].

## Data availability
The data that support the findings of this study are available from the corresponding author upon reasonable request.

## Code availability
The package we used for data cleaning and analysis is provided in a public repository. It is available at https://github.com/gasfas/ANACONDA_2.git.

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

## Acknowledgements

The authors would like to thank Joakim Laksman, Rollin Maxence, Erik Månsson, Maxim Tchaplyguine, Gunnar Öhrwall and other Max-lab staff for their help during the experiments. We are profoundly grateful to Francisco González-Galindo, Antoine Martinez and Jean-Yves Chaufray for their insightful comments about the possible implications of our study to the CO$_2$-rich atmospheres. This article is based upon work from COST action CA18212—Molecular Dynamics in the GAS phase (MD-GAS), supported by COST (European Cooperation in Science and Technology). The authors acknowledge the generous allocation of computer time at the Centro de Computación Científica at the Universidad Autónoma de Madrid (CCC-UAM). This work was partially supported by the MICINN (Spanish Ministry of Science and Innovation) project PID2019-110091GB-I00, funded by MCIN/AEI/10.13039/501100011033 and the 'María de Maeztu' (CEX2018-000805-M) Programme for Centres of Excellence in R&D. D.B. acknowledges the FPI grant associated with the MICINN project CTQ2016-76061-P. We also acknowledge the support of the Helmholtz Foundation through the Helmholtz-Lund International Graduate School (HELIOS, HIRS-0018) and the Swedish Research Council (VR2020-0520).

## Author contributions

M.G., N.W. and S.L.S. performed the measurements, S.G. processed the experimental data and performed the analysis with aid from B.O. under the supervision of M.G. D.B.L. and S.D.T. performed the theoretical calculations. S.G., D.B.L., M.G., S.D.T., S.L.S. and N.W. wrote the paper. S.G. and D.B.L. designed the figures with input from all authors. All authors provided critical feedback and participated in scientific discussions.

## Funding

## Competing interests

The authors declare no competing interests.
