## [Peer Review File · Communications Chemistry]

Reviewers' comments:

Reviewer #1 (Remarks to the Author):

The article by Smita Ganguly et al. titled "The origin of enhanced O₂⁺ production from photoionized CO₂ clusters" provides a clear mechanism for the formation of O₂⁺ from CO₂ clusters, which is much faster than from molecular CO₂. The article is well written and various methods were used to reach the same conclusions. I have only minor corrections/additions to the manuscript:

Reading the article, I was wondering about two things that should be mentioned somewhere in the introduction already. First, why do you expect the production of O₂⁺ be more than what could be explained by CO₂ molecule? Is a lot of O₂⁺ seen in the atmosphere of Mars and a new mechanism was needed to explain observations? Second, why are the clusters doubly charged? This is partially explained in the the discussion of Fig. 4 when you say that the second electron is likely lost after the first one, but it comes quite late in the article and it doesn't explain why you didn't look into the singly charged clusters. Presumably because the experiments showed that high amount of O₂⁺ is formed in the doubly charged case?

Page 7, line 94: You discuss the neutral fragment of the cluster and how its momentum is calculated. How can you be sure that there is only one neutral fragment in addition to the two charged fragments?

Page 13, line 182: Aren't there also smaller clusters in the mixture that can form O₂⁺? When the mean cluster size increases, you probably also have less of the smallest clusters and more in the 10-44 CO₂ range that are producing more O₂⁺.

Page 14, line 215: This should say "described by Laksman et al.²⁶", right?

Page 15, line 229: Do you mean that you used Takeuchi's method to create the structures (which it sounds like) or you took the structures from their article's SI? Also, the paper has no "et al."

In the main text, add more specific references to the SI. The SI file is quite long, so finding the correct sections is difficult.

It would be beneficial to say clearly whether each your results come from experiments or calculations, since you have done both in the study. For example, in Fig. 5 the markers are presumably the experiments, but where does the stable cluster size $N_{\text{mean}}=44$ comes from?

Is there a particular reason why the stationary points on the Potential Energy Surfaces weren't calculated at a higher level of theory? CCSD(T) perhaps. The smallest clusters at least aren't too large for some coupled cluster methods, such as F12 and DLPNO.

In Supplement:

Page 4: "Cluster fragments larger than (CO₂)_n were not observed, possibly due to the low detection efficiency of the heavy ions." Why is the detection efficiency of the larger clusters low? Do you think it's because of low transmission or something else?

Page 4: You talk about regions A and B indicated in Fig. S.7. However, I don't see any mention of

regions or A and B in Fig. S.7. Do you mean Fig. 2 in the main text?

Page 10, 4th line: What does "therefore being their favourable formation." mean?

End of page 10: "Figure S.8 shows the NBO charge analysis of $(\text{CO}_2)_{42}^+$, $(\text{CO}_2)_{82}^+$ and $(\text{CO}_2)_{132}^+$. Also, is it possible to change the color scale in Fig. S.8 so that the bright red color would match the charge value of the minimum charge in the shown cases? Now it's not that easy to see the difference between different oxygen atoms because the colors are so similar.

Page 14, 6th line: What do you mean by "concerted mechanism"? That both reaction paths are happening simultaneously?

Reviewer #2 (Remarks to the Author):

The authors report combined soft X-ray ionization experiments and quantum chemistry calculation on the photodissociation of CO_2 clusters. Enhanced O_2^+ productions from photoionized CO_2 clusters were founded and the reaction pathways were calculated. The size dependent effect for the production of O_2^+ was observed, which are very difficult to record. The work is convincing and could be published after the authors address a single question: The authors described that the yields of O_2^+ increase while the size of CO_2 clusters in the small sizes, then decrease to a constant level for larger clusters. However, Fig 5 shows that the O_2^+ yields have a significant decrease near the cluster size of ~ 100 (both total O_2^+ and $\text{O}_2^+/\text{CO}_2^+$). Is this because of experimental error or the yields decrease for the extremely larger clusters? The authors should provide some comments.

Reviewer #3 (Remarks to the Author):

The article "The origin of enhanced O_2^+ production from photoionized CO_2 clusters" by Ganguly et al. reports on an experimental/theoretical co-study on photo-induced rearrangement of small to medium-sized CO_2 clusters. The work is motivated by a surprisingly large abundance of O_2 in (parts of) the atmosphere of Mars and proposes an O_2 -formation occurring in doubly ionized CO_2 clusters. This formation process does not exist in isolated CO_2 monomers. The article is clearly written and nicely accessible and the presented data and findings seem valid and sound.

The interpretation of experiments on cluster beams are particularly challenging and I am delighted to see the level of detail on the occurring processes that the authors were able to obtain from comparison of their beautiful experiment to the presented state-of-the-art modelling. I believe the findings presented here shed new light on the topic and highlight the importance of a "chemical environment" which is often ignored. I support a publication of the article in principle as is. I'd like, however, the authors to briefly consider the following points:

A representation as a Dalitz plot is typically hard to access for non-experts. Due to conservation laws, however, different regions of the plot correspond to distinct geometrical arrangements of the three plotted "particles" in position space. Maybe the authors find a way to decode their Dalitz plots with respect to this in a way which is easier to understand for a broader audience?

Determining the cluster size has been done by the Gamma^* -approach. How valid is the approach

here and how reliable is the extracted cluster size information? I assume if the real cluster sizes differed from the extracted ones, the overall interpretation would still be valid? (Or do the simulations indicate that this assumption is not true?)

Again, very nice and interesting work!

We would like to thank all referees for their critical comments and the valuable appreciation of our work. In the following we provide a point-to-point answer to the referees.

Reviewer 1

Comment 1. The article by Smita Ganguly et al. titled "The origin of enhanced O_2^+ production from photoionized CO_2 clusters" provides a clear mechanism for the formation of O_2^+ from CO_2 clusters, which is much faster than from molecular CO_2 . The article is well written and various methods were used to reach the same conclusions. I have only minor corrections/additions to the manuscript:

We would like to thank the reviewer for the positive evaluation of our work, and for the valuable comments, which we address below:

Comment 2. Reading the article, I was wondering about two things that should be mentioned somewhere in the introduction already. First, why do you expect the production of O_2^+ be more than what could be explained by CO_2 molecule? Is a lot of O_2^+ seen in the atmosphere of Mars and a new mechanism was needed to explain observations?

Yes, space probe measurements show that O_2^+ is the dominant ion in the Martian ionosphere. There are various photochemical models of the Martian ionosphere that estimate the ionic composition. In the upper ionosphere (altitudes >100 km), the amount of O_2^+ can be modelled. It is due to ion-molecule collision reactions of O^+ with CO_2 molecules. However, at the interface to the mesosphere (altitudes ~70 km), an excess of O_2^+ is observed which is not explained by the current model. Therefore, a new mechanism with high O_2^+ yield is highly relevant. Changes have been made in the Introduction section lines 24-33 and are highlighted in red.

Comment 3. Second, why are the clusters doubly charged? This is partially explained in the discussion of Fig. 4 when you say that the second electron is likely lost after the first one, but it comes quite late in the article, and it doesn't explain why you didn't look into the singly charged clusters. Presumably because the experiments showed that high amount of O_2^+ is formed in the doubly charged case?

The reason is explained in line 33 and we think it fits well in the Introduction. Since ionization of the C1s electrons leads to rapid Auger decay producing doubly charged clusters we do not expect to have any singly charged clusters, and therefore our study focuses on dication species. It is known from electron impact ionization that O_2^+ is produced even for low kinetic energy collision (~40-50 eV), see refs 25-27.

Comment 4. Page 7, line 94: You discuss the neutral fragment of the cluster and how its momentum is calculated. How can you be sure that there is only one neutral fragment in addition to the two charged fragments?

We agree that there can be more than one undetected fragment. The model does not make any assumptions on the number or charge of undetected fragments. We define the

residual momentum as the sum of individual momenta of all the undetected fragments. Experimentally, we calculate this residual momentum from the detected fragments using the law of conservation of momentum. We then depict the momentum correlation between the detected fragments and residual momentum in the Dalitz plot. Our conclusion that the dissociation mechanisms are sequential will also hold for multiple undetected fragments.

Comment 5. Page 13, line 182: Aren't there also smaller clusters in the mixture that can form O_2^+ ? When the mean cluster size increases, you probably also have less of the smallest clusters and more in the 10-44 CO_2 range that are producing more O_2^+ .

As we tune the pressure and increase the mean cluster size (which follows a log-normal distribution) the population of smaller clusters will decrease. So, at mean cluster sizes larger than 40, the contribution from the dimers and trimers will be negligible.

Comment 6. Page 14, line 215: This should say "described by Laksman et al.²⁶", right?

The referee is correct, changes are made in text (highlighted in red).

Comment 7. Page 15, line 229: Do you mean that you used Takeuchi's method to create the structures (which it sounds like) or you took the structures from their article's SI? Also, the paper has no "et al."

The referee is correct, we have taken the structures directly from Takeuchi's article Supporting Information. Following the referee's suggestion, we have clarified this point in the main text of the revised version, as well as removed the 'et al.'. Thank you for pointing this out.

Comment 8. In the main text, add more specific references to the SI. The SI file is quite long, so finding the correct sections is difficult.

Specific references have been added to the main text and highlighted in red.

Comment 9. It would be beneficial to say clearly whether each your results come from experiments or calculations, since you have done both in the study. For example, in Fig. 5 the markers are presumably the experiments, but where does the stable cluster size $N_{mean}=44$ comes from?

The caption now clearly states that these are experimental results, as highlighted in red. The stable cluster sizes of multiply charged CO_2 clusters was reported in ref 22 by analysing the mass spectrum at different ionization energies.

Comment 10. Is there a particular reason why the stationary points on the Potential Energy Surfaces weren't calculated at a higher level of theory? CCSD(T) perhaps. The smallest clusters at least aren't too large for some coupled cluster methods, such as F12 and DLPNO.

This is an interesting point. The computed relative energies at the DFT level of theory are expected to provide accurate results with a relatively cheap computational effort. Nevertheless, to check and proof this accuracy, we have performed calculations using the level of theory suggested by the reviewer, DLNPO-CCSD(T) def2-TZVP, for one of the proposed mechanisms. These results have been added in Figure S9 in the revised S.I. and show a very good correspondence for the energetic values between both CC and DFT methods. Therefore, we can be confident of the level of theory used to compute the rest of the mechanisms.

Comment 11. In Supplement: Page 4: "Cluster fragments larger than $(\text{CO}_2)^+_{10}$ were not observed, possibly due to the low detection efficiency of the heavy ions." Why is the detection efficiency of the larger clusters low? Do you think it's because of low transmission or something else?

In our data set, fragments in coincidence are limited to cluster sizes below $(\text{CO}_2)^+_{10}$. To avoid confusion, we have changed the text and added discussion about the probabilities of detection in S.I. and highlighted in red. In single coincidence (only one ion measured within the acquisition time window after photoelectron trigger), we observed much larger cluster fragments. The measured signal for large cluster fragments can be affected by two factors: lower detection efficiency for heavy ions or a decreased transmission of the spectrometer. The detection efficiency decreases as a function of the mass of the fragment, which has its origin in the probability of converting an ion into an electron by the microchannel plate detector - see Gilmore et al Int. J. Mass. Spectro 202, 217 (2000). For our experimental conditions, with 4 keV ion kinetic energy, we detect signals from ions with mass up to $(\text{CO}_2)^+_{30}$ in addition to doubly-charged species $(\text{CO}_2)^{2+}_{44-47}$. The decrease of the transmission efficiency can occur when the transverse cluster velocity in the supersonic beam is high. Hence, after a long time of flight, the transverse drift of the fragments can be larger than the radius of the detector or even the drift tube. In our experimental conditions, with a transverse jet velocity of ~ 680 m/s, the decrease of the transmission efficiency is not relevant. Therefore, the detection efficiency can be a limiting factor for larger cluster fragments. In fig 5, we account for it by considering all possible O_2^+ events, and our conclusions are valid regardless of the detection efficiency.

Comment 12. Page 4: You talk about regions A and B indicated in Fig. S.7. However, I don't see any mention of regions or A and B in Fig. S.7. Do you mean Fig. 2 in the main text?

Yes, we are referring to the Fig 2. In the main text. Changes have been made in text and are highlighted in red.

Comment 13. Page 10, 4th line: What does "therefore being their favourable formation." mean?

We thank the referee for pointing out this typo. We meant that when analysing the computed adiabatic ionization potentials, AIPs, for the 2Y structures their formation turns out to be thermodynamically favourable. We have rewritten this sentence in the revised version to clarify this point.

Comment 14. End of page 10: "Figure S.8 shows the NBO charge analysis of $(\text{CO}_2)^{2+}_4$ ", $(\text{CO}_2)^{2+}_8$ and $(\text{CO}_2)^{2+}_{13}$. Also, is it possible to change the color scale in Fig. S.8 so that the bright red color would match the charge value of the minimum charge in the shown cases? Now it's not that easy to see the difference between different oxygen atoms because the colors are so similar.

Indeed, the colour range was difficult to distinguish in the previous version. We have followed the referee's suggestion and set the red bright colour as the minimum charge value in the atoms. We have updated Figure S8. in the S.I. and with this new version the difference in atomic charges is clearly visible.

Comment 15. Page 14, 6th line: What do you mean by "concerted mechanism"? That both reaction paths are happening simultaneously?

The referee is correct; As she/he suggested, the production of CO_2^+ and O_2^+ through this mechanism can simultaneously occur from the fragmentation of CO_2CO^+ and CO_2CO_3^+ respectively. We have explained this mechanism more carefully to avoid confusion in the revised version.

Reviewer 2

Comment 1. The authors report combined soft X-ray ionization experiments and quantum chemistry calculation on the photodissociation of CO₂ clusters. Enhanced O₂⁺ productions from photoionized CO₂ clusters were founded and the reaction pathways were calculated. The size dependent effect for the production of O₂⁺ was observed, which are very difficult to record. The work is convincing and could be published after the authors address a single question:

We would like to thank the reviewer for the positive evaluation of our work and for the valuable comment which we address below:

Comment 2. The authors described that the yields of O₂⁺ increase while the size of CO₂ clusters in the small sizes, then decrease to a constant level for larger clusters. However, Fig 5 shows that the O₂⁺ yields have a significant decrease near the cluster size of ~100 (both total O₂⁺ and O₂⁺/CO₂⁺). Is this because of experimental error or the yields decrease for the extremely larger clusters? The authors should provide some comments.

We believe that the decrease in yield of O₂⁺ near the cluster size of ~100 is due to lower statistics and hence experimental error. In the supplementary information in figure S.4. we show results from a data set for cluster sizes up to 400 where the yield does not exhibit a decrease for larger cluster sizes.

Reviewer 3

Comment 1. The article “The origin of enhanced O_2^+ production from photoionized CO_2 clusters” by Ganguly et al. reports on an experimental/theoretical co-study on photo-induced rearrangement of small to medium-sized CO_2 clusters. The work is motivated by a surprisingly large abundance of O_2 in (parts of) the atmosphere of Mars and proposes an O_2 -formation occurring in doubly ionized CO_2 clusters. This formation process does not exist in isolated CO_2 monomers. The article is clearly written and nicely accessible and the presented data and findings seem valid and sound. The interpretation of experiments on cluster beams are particularly challenging and I am delighted to see the level of detail on the occurring processes that the authors were able to obtain from comparison of their beautiful experiment to the presented state-of-the-art modelling. I believe the findings presented here shed new light on the topic and highlight the importance of a “chemical environment” which is often ignored. I support a publication of the article in principle as is. I’d like, however, the authors to briefly consider the following points:

We would like to thank the reviewer for the positive evaluation of our work and for the valuable comments which we address below:

Comment 2. A representation as a Dalitz plot is typically hard to access for non-experts. Due to conservation laws, however, different regions of the plot correspond to distinct geometrical arrangements of the three plotted “particles” in position space. Maybe the authors find a way to decode their Dalitz plots with respect to this in a way which is easier to understand for a broader audience?

We have tried to simplify a multi-body cluster problem to a three-body problem to make the interpretation more accessible. Therefore, geometric arrangements, such as Newton plots traditionally used for the molecular breakup, can be misleading, so we prefer to use a momentum-sharing plot such as the Dalitz plot where signature of the two processes is visually straightforward. While a deeper explanation of the process is available via the Dalitz plot, this is not necessary for the reader to appreciate our results. For, those interested to reflect on the momentum correlation we have added a guide to read the Dalitz plot in SI (Fig S.2).

Comment 3. Determining the cluster size has been done by the Gamma-approach. How valid is the approach here and how reliable is the extracted cluster size information? I assume if the real cluster sizes differed from the extracted ones, the overall interpretation would still be valid? (Or do the simulations indicate that this assumption is not true?)*

The Gamma*-approach we have used has been verified by Harnes et al. (2011) with photoelectron spectroscopy on clusters produced with the same cluster source. This cluster source has also been tested on a large variety of systems, see Björneholm, O. *et al* 601, 161 NIMA (2009) and references therein. However, the extracted cluster sizes do have finite error bars. Even if the real cluster sizes differed significantly from the extracted ones, our overall interpretation would still be valid. The trend increased O_2^+ production for larger cluster sizes remains unchanged.

Comment 4. Again, very nice and interesting work!

Thank you!

REVIEWERS' COMMENTS:

Reviewer #1 (Remarks to the Author):

The authors have addressed all of my comments in their revised manuscript. I recommend that the manuscript is published as is, with no further review.

I have only one clarification for the first point of "Comment 14". Maybe the authors missed it, I see I didn't write it very clearly. The text in the Supporting information only mentions clusters 4 and 8, but Figure S.8 also has cluster 13.

Reviewer #2 (Remarks to the Author):

My questions have been fully addressed in the revised manuscript. I suggest this work to publish as it is.

Reviewer #3 (Remarks to the Author):

The authors have addressed all my concerns.

We would like to thank all referees for their critical comments and the valuable appreciation of our work. In the following we provide a point-to-point answer to the referees.

Reviewer 1

Comment 1. The authors have addressed all of my comments in their revised manuscript. I recommend that the manuscript is published as is, with no further review.

We would like to thank the reviewer for their valuable comment.

Comment 2. I have only one clarification for the first point of "Comment 14". Maybe the authors missed it, I see I didn't write it very clearly. The text in the Supporting information only mentions clusters 4 and 8, but Figure S.8 also has cluster 13.

We apologise for missing this comment in the previous revision. We have now added cluster 13 in the text of the Supporting Information.